# The Effect of Isoleucine Supplementation on Body Weight Gain and Blood Glucose Response in Lean and Obese Mice

**DOI:** 10.3390/nu12082446

**Published:** 2020-08-14

**Authors:** Rebecca O’Rielly, Hui Li, See Meng Lim, Roger Yazbeck, Stamatiki Kritas, Sina S. Ullrich, Christine Feinle-Bisset, Leonie Heilbronn, Amanda J. Page

**Affiliations:** 1Adelaide Medical School, University of Adelaide, Adelaide, SA 5005, Australia; rebecca.orielly@adelaide.edu.au (R.O.); hui.li01@adelaide.edu.au (H.L.); sina.s.ullrich@gmail.com (S.S.U.); christine.feinle@adelaide.edu.au (C.F.-B.); leonie.heilbronn@adelaide.edu.au (L.H.); 2Nutrition, Diabetes and Gut Health, Lifelong Health Theme, South Australian Health and Medical Research Institute (SAHMRI), Adelaide, SA 5001, Australia; 3School of Agriculture, Food and Wine, The University of Adelaide, Glen Osmond, SA 5064, Australia; seemeng.lim@adelaide.edu.au; 4Centre for Community Health Studies, Faculty of Health Sciences, Universiti Kebangsaan Malaysia, Kuala Lumpur 50300, Malaysia; 5College of Medicine and Public Health, Flinders Medical Centre, Flinders University, Bedford Park, SA 5042, Australia; roger.yazbek@flinders.edu.au; 6Women’s and Children’s Hospital, North Adelaide, SA 5006, Australia; stamatiki.kritas@gmail.com; 7Clinical Trial Unit, Department of Clinical Research, University of Basel and University Hospital Basel, 4031 Basel, Switzerland

**Keywords:** obesity, amino acid, isoleucine, chronic supplementation, energy expenditure, oral glucose tolerance test, glycaemic control, gastric emptying breath test

## Abstract

Chronic isoleucine supplementation prevents diet-induced weight gain in rodents. Acute-isoleucine administration improves glucose tolerance in rodents and reduces postprandial glucose levels in humans. However, the effect of chronic-isoleucine supplementation on body weight and glucose tolerance in obesity is unknown. This study aimed to investigate the impact of chronic isoleucine on body weight gain and glucose tolerance in lean and high-fat-diet (HFD) induced-obese mice. Male C57BL/6-mice, fed a standard-laboratory-diet (SLD) or HFD for 12 weeks, were randomly allocated to: (1) Control: Drinking water; (2) Acute: Drinking water with a gavage of isoleucine (300 mg/kg) prior to the oral-glucose-tolerance-test (OGTT) or gastric-emptying-breath-test (GEBT); (3) Chronic: Drinking water with 1.5% isoleucine, for a further six weeks. At 16 weeks, an OGTT and GEBT was performed and at 17 weeks metabolic monitoring. In SLD- and HFD-mice, there was no difference in body weight, fat mass, and plasma lipid profiles between isoleucine treatment groups. Acute-isoleucine did not improve glucose tolerance in SLD- or HFD-mice. Chronic-isoleucine impaired glucose tolerance in SLD-mice. There was no difference in gastric emptying between any groups. Chronic-isoleucine did not alter energy intake, energy expenditure, or respiratory quotient in SLD- or HFD-mice. In conclusion, chronic isoleucine supplementation may not be an effective treatment for obesity or glucose intolerance.

## 1. Introduction

The branched-chain amino acids (BCAAs), isoleucine, leucine and valine, are essential amino acids accounting for ~35% of the essential amino acids comprising muscle proteins in humans and ~40% of the pre-formed amino acids required by all mammals [1]. In population studies, an elevated dietary intake of BCAAs was associated with a lower prevalence of overweight and obesity in adults [2,3]. Further, BCAA supplementation was demonstrated to preserve lean muscle mass during weight loss [4,5,6]. This evidence suggests a role for BCAA supplementation in the treatment of obesity. In particular, chronic isoleucine supplementation in rodents has been demonstrated to prevent high-fat diet (HFD)-induced obesity [7]. However, whether chronic isoleucine supplementation is an effective approach to ameliorate weight gain and promote weight loss in established obesity, is unknown. 

Acute administration of isoleucine and leucine in rats improved glucose tolerance, with isoleucine showing greater effectiveness than leucine [8]. This was attributed to the synergistic action of isoleucine with endogenous insulin to enhance glucose uptake into tissues [9,10,11]. In addition, acute isoleucine supplementation improved glucose tolerance in leptin receptor-deficient (*db/db)* mice, a model of morbid obesity and hyperglycaemia [12]. This evidence suggests that isoleucine supplementation may be useful in the treatment of glucose intolerance. However, whether this glucose-lowering effect persists following a chronic supplementation regime is unknown.

It is known that postprandial blood glucose levels are influenced by the rate of gastric emptying [13]. In particpants with type 2 diabetes, consumption of whey protein before a meal slowed gastric emptying and was associated with lower postprandial blood glucose levels [14,15]. Further, in healthy lean participants, acute intragastric administration of isoleucine lowered the blood glucose response to a mixed nutrient drink, which was attributed to a slowing of gastric emptying [16]. Therefore, we hypothesised that chronic isoleucine supplementation will slow gastric emptying, improve glucose tolerance, and reduce body weight gain in mice.

The current study aimed to determine whether chronic dietary supplementation with the BCAA isoleucine, alters body weight gain, adiposity, glucose tolerance, and energy metabolism in mice with HFD-induced obesity.

## 2. Materials and Methods

### 2.1. Ethics Approval

This study was approved (Ethics approval: SAM237) by the South Australian Health and Medical Research Institute Animal Ethics Committee. All experimental protocols were performed in alignment with the Australian Code of Practice for the Care and Use of Animals for Scientific Purposes.

### 2.2. Study Design

Eight-week-old male C57BL/6 mice (*n* = 54) were group-housed in a 12:12 h light-dark cycle within a temperature (24 ± 1 °C) controlled facility. Mice were provided ad libitum access to either a standard laboratory diet (SLD; 12%, 23%, and 65% of energy from fat, protein, and carbohydrates, respectively; Specialty Feeds, Western Australia, Australia; *n* = 30) or HFD (60%, 20%, and 20% of energy from fat, protein, and carbohydrates, respectively; adapted from Research Diets Inc., New Brunswick, NJ, USA; *n* = 24). Consistent with the previous literature, the HFD-induced obese mouse model was chosen as both a model of obesity [17] and impaired glucose tolerance [18]. After 12 weeks on their respective diets, a sub-group of SLD (*n* = 10; SLD-Chronic (Ch)) and HFD-mice (*n* = 8; HFD-Ch) received ad libitum isoleucine (1.5% *w*/*v*; Purebulk Inc., Roseburg, OR, USA) supplemented in the drinking water. The remaining SLD (*n* = 20) and HFD-mice (*n* = 16) continued with ad libitum access to normal drinking water. At 16 weeks, all mice were singly housed and underwent an OGTT (at 1400 h) and gastric emptying breath test (at 0900 h) in random order with a three-day recovery between tests. In each diet group, the mice provided normal drinking water were subdivided into two groups, receiving either an oral gavage of water (*n* = 10, SLD-Control (C); *n* = 8, HFD-C) or isoleucine (300 mg/kg body weight; *n* = 10, SLD-Acute (A); *n* = 8, HFD-A) 30 min before the OGTT or GEBT. The SLD and HFD-Ch mice received an oral gavage of drinking water similar to the control groups. The doses for acute and chronic isoleucine treatments were chosen based on previous studies [8,12]. At 17 weeks, all mice were placed in metabolic monitoring cages. The body weight of all the mice was measured weekly, except the final two weeks due to the different interventions. 

### 2.3. Oral Glucose Tolerance Test

Consistent with previous studies [19,20], mice were fasted for six hours (0800–1400 h) before receiving an oral gavage of either isoleucine (SLD/HFD-A groups) or water (SLD/HFD-C and SLD/HFD-Ch groups). After 30 min, all mice received an oral gavage of 20% D-glucose (1 g/kg BW), a dose chosen to ensure the HFD-mice did not experience a severe hyperglycemic response with blood glucose levels beyond the range of the glucose monitor. Blood was collected from a tail prick before isoleucine/water administration, considered as the baseline, and again at 15, 30, 45, 60, and 120 min post glucose administration. Blood glucose levels were determined with an ACCU CHEK Performa monitor (ACCU CHEK, New South Wales, Australia). 

### 2.4. Gastric Emptying Breath Test

Gastric emptying of a solid meal was determined using a non-invasive breath test as previously described [21,22]. Mice were fasted overnight (1600—0900 h) prior to an oral gavage of either isoleucine (SLD/HFD-A) or water (SLD/HFD-C and SLD/HFD-Ch). After 30 min, all mice were provided 0.1 g of baked egg yolk containing ^13^C-octanoic acid (1 μL/g; 99% enrichment, Cambridge Isotope Laboratories, Andover, MA, USA) to consume voluntarily within 1 min. Breath samples were collected before isoleucine administration (baseline; 0 min) and again at regular intervals (5 min intervals from 5–30 min and 15-min intervals from 30–150 min) after egg consumption. Breath samples were analysed for the ^13^CO_2_ content using an isotope ratio mass spectrometer (Europa Scientific, Crewe, UK). The ^13^CO_2_ excretion data were analysed by non-linear regression analysis for curve fitting and for calculation of gastric half emptying time (t ½) [23]. Gastric half emptying time was not measured in the HFD-mice due to sampling difficulties; the egg yolk was not consumed within 1 min which invalidates results.

### 2.5. Metabolic Monitoring

Mice were individually housed in Promethium metabolic cages (Sable Systems International, North Las Vegas, NV, USA) for 72 h of continuous metabolic monitoring. Energy intake (kJ), energy expenditure (kJ/lean mass), respiratory quotient (RQ; VCO_2_/VO_2_), and total activity (meters, m) were measured and analysed using the ExpeData data analysis software (Sable Systems International, North Las Vegas, NV, USA).

### 2.6. Tissue Collection

Mice were fasted overnight (1600–0900 h) then anaesthetised with isoflurane (5% in medical oxygen). The nose-to-tail length and abdominal circumference of mice were measured. Blood was collected from the abdominal aorta and transferred to ethylenediaminetetraacetic acid (EDTA) tubes (Thermo Fisher Scientific, Victoria, Australia). Plasma was extracted by centrifugation at 1000 g and 4 °C for 15 min, and snap-frozen in liquid nitrogen prior to storage at −80 °C until further analysis. Liver, gonadal fat pads, and inter-scapula brown fat pads were collected and weighed. Lean mass was determined by the final body weight minus the weight of collected fat pads. A section of the liver was fixed in 4% paraformaldehyde for 4 h, cryoprotected overnight in 30% sucrose in a phosphate buffer, frozen in Tissue-Tek O.C.T. compound (Sakura Finetek USA Inc., Torrance, CA, USA) and stored at −80 °C before processing for histology. 

### 2.7. Plasma Metabolites

Plasma total triglycerides, total cholesterol, and high-density lipoprotein (HDL)-cholesterol concentrations were measured using commercial enzymatic kits (OSR60118, OSR6116, and OSR6187, respectively (Beckman Coulter Inc., Georgia, USA)) on a Beckman AU480 clinical analyser (Beckman Coulter Inc., Atlanta, GA, USA). Plasma low-density lipoprotein (LDL)-cholesterol concentrations were estimated using the Friedewald equation [24]: LDL-cholesterol = Total cholesterol − (total triglyceride /2.2) − HDL-cholesterol

### 2.8. Liver Lipid Content 

The histological lysochrome lipid stain Oil Red O was performed on liver sections using a standard protocol [25]. Slides were imaged using a NanoZoomer digital slide scanner (Hamamatsu Photonics, Hamamatsu City, Japan) and analysed for the average percentage of stained lipid per 1 mm^2^ area using the ImageJ-win64 software.

### 2.9. Statistical Tests

Results are expressed as the mean ± SEM. A two-way ANOVA was performed to assess diet and isoleucine treatment effects with a Tukey’s post hoc test for multiple comparisons, using the GraphPad Prism v8 software (GraphPad, California, USA). The OGTT_0–120 min_ blood glucose area under the curve (AUC) was generated using the IBM SPSS Statistics 26 software (IBM, New York, NY, USA). A correlation between gastric half emptying time (t ½) and OGTT_0–120 min_ blood glucose AUC for SLD groups was performed in the GraphPad Prism v8 software. The coefficient of determination value (r^2^) was considered significant at *p* < 0.05.

## 3. Results

### 3.1. Chronic Isoleucine Treatment Does Not Affect Weight Gain and Adiposity

At the beginning of week 12, prior to isoleucine supplementation, HFD-mice gained significantly more weight than SLD-mice (*p* < 0.001, unpaired *t*-test; Figure 1A). There was no difference in body weight between different treatment groups in mice fed a SLD or HFD prior to chronic isoleucine treatment (*p* < 0.0001, *F* (1, 50) = 28.17, diet effect; *p* = 0.1262, *F* (1, 50) = 2.418, isoleucine effect; SLD-C/A; 34.8 ± 0.7 g (*n* = 20), SLD-Ch; 37.2 ± 1.3 g (*n* = 10), HFD-C/A; 42.3 ± 1.6 g (*n* = 16), and HFD-Ch; 44.1 ± 1.7 g (*n* = 8)).

In weeks 12–15, HFD-mice continued to gain more weight compared to SLD-mice (*p* < 0.0001, *F* (1, 50) = 19.21, diet effect; Figure 1A), but there was no effect of chronic isoleucine treatment on weight gain (Figure 1A).

At week 18, abdominal circumference was greater in HFD-mice than SLD-mice, but was not affected by chronic isoleucine treatment (*p* < 0.0001, *F* (1, 26) = 24.78, diet effect; SLD-C, 9.3 ± 0.2 cm, SLD-Ch, 9.9 ± 0.5 cm, HFD-C, 11.4 ± 0.3 cm, and HFD-Ch, 11.2 ± 0.3 cm). In addition, HFD-mice had heavier gonadal fat pads and brown fat pads than SLD-mice (both *p* < 0.01, *F* (1, 50) = 11.13 and *F* (1, 47) = 11.44, respectively, diet effect; Figure 1(Bi,Bii)), but these parameters were not affected by the chronic isoleucine treatment.

### 3.2. Chronic Isoleucine Treatment Does Not Affect Liver Lipid Content

HFD-mice had heavier livers than SLD-mice (*p* < 0.0001, *F* (1, 50) = 18.34, diet effect; Figure 2A). The liver lipid content was greater in HFD-mice than SLD-mice (*p* < 0.0001, *F* (1, 28) = 37.31, diet effect; Figure 2B). There was no effect of the chronic isoleucine treatment on liver mass or lipid content (Figure 2A,B).

### 3.3. Chronic Isoleucine Treatment Does Not Alter Energy Intake, Energy Expenditure, Activity, and Respiratory Quotient

HFD-mice consumed more energy across 24 h (*p* < 0.05, *F* (1, 50) = 4.157, diet effect; Figure 3(Ai)) compared to SLD-mice, predominantly due to increased energy intake during the light phase (*p* < 0.05, *F* (1, 50) = 5.715, diet effect; Figure 3(Aii)). There was no effect of the chronic isoleucine treatment on total energy intake across 24 h, during the light phase or dark phase (Figure 3(Ai–Aiii)). HFD feeding and chronic isoleucine treatment had no effect on 24 h of total water intake (SLD-C; 3.5 ± 0.07 mL/day, SLD-Ch; 3.8 ± 0.17 mL/day, HFD-C; 3.5 ± 0.1 mL/day, and HFD-Ch; 3.6 ± 0.2 mL/day).

HFD-mice had a significantly lower energy expenditure (normalised to lean body mass) compared to SLD-mice across 24 h, during the light phase or dark phase (*p* < 0.01, *F* (1, 50) = 9.151, *p* < 0.05, *F* (1, 50) = 6.499, *p* < 0.01, *F* (1, 50) = 11.25, respectively, diet effect; Figure 3(Bi–Biii)). A significant diet by the chronic isoleucine treatment interaction was observed in energy expenditure across 24 h (*p* < 0.05, *F* (1, 50) = 5.416, interaction; Figure 3(Bi)) and during the dark phase (*p* < 0.05, *F* (1, 50) = 6.468, interaction; Figure 3(Biii)). During the dark phase, the chronic isoleucine treatment reduced energy expenditure in SLD-mice (*p* < 0.05, Sidak’s post hoc test), but not in HFD-mice.

Total activity levels were not affected by HFD feeding or chronic isoleucine treatment across 24 h or during the light or dark phase (Figure 3(Ci–Ciii)).

HFD-mice had lower RQ values compared to SLD-mice across 24 h (*p* < 0.001, *F* (1, 50) = 19.66, diet effect; Figure 3(Ci)), during the light phase (*p* < 0.05, *F* (1, 50) = 4.082, diet effect; Figure 3(Cii)) and dark phase (*p* < 0.001, *F* (1, 50) = 38.4, diet effect; Figure 3(Ciii)). Average RQ values were not affected by the chronic isoleucine treatment across 24 h, during the light phase or dark phase (Figure 3(Ci–Ciii)).

### 3.4. Chronic Isoleucine Treatment Does Not Affect Plasma Lipid Metabolites

HFD-mice had elevated plasma total triglycerides (*p* < 0.001, *F* (1, 30) = 31.28, diet effect; Figure 4A), total cholesterol (*p* < 0.001, *F* (1, 30) = 40.02, diet effect; Figure 4B), HDL-cholesterol (*p* < 0.001, *F* (1, 30) = 40.16, diet effect; Figure 4C), and LDL-cholesterol (*p* < 0.0001, *F* (1, 30) = 25.36, diet effect; Figure 4D) compared to SLD-mice. There was no effect of chronic isoleucine treatment on these plasma lipid metabolites (Figure 4A–D).

### 3.5. Acute and Chronic Isoleucine Treatment Differentially Affect Glucose Tolerance in SLD- and HFD-Mice

HFD-mice had higher fasting blood glucose levels than SLD-mice (*p* < 0.0001, *F* (1, 48) = 19.34, diet effect; Figure 5(Ai,Bi)). There was no effect of the chronic isoleucine treatment on fasting blood glucose levels (Figure 5(Ai,Bi)).

HFD-mice had a greater glucose AUC than SLD-mice (*p* < 0.001, F (1, 48) = 34.44, diet effect; Figure 5(Aii,Bii)). In SLD groups, an elevated glucose AUC was observed in chronic isoleucine treated mice compared to control mice (*p* < 0.05, one-way ANOVA; Figure 5(Aii)), but there was no difference between acute and chronic isoleucine treated mice. In HFD groups, there was no effect of acute or chronic isoleucine treatment on glucose AUC (Figure 5(Bii)).

### 3.6. Acute and Chronic Isoleucine Treatment Do Not Affect Gastric Emptying

There was no significant difference in the gastric half emptying time (t ½) between different isoleucine groups in SLD-mice (SLD-C, 125.3 ± 13.2 min (*n* = 8); SLD-A, 144.4 ± 15.3 min (*n* = 8) and SLD-Ch, 126.7 ± 5.1 min (*n* = 7)).

There was no correlation between gastric half emptying time and blood glucose AUC in SLD-mice (correlation coefficient of determination value (r^2^); SLD-C, r^2^ = 0.0197, SLD-A, r^2^ = 0.2602, and SLD-Ch, r^2^ = 0.004292).

## 4. Discussion

Longitudinal population studies have demonstrated an inverse association between dietary BCAA consumption and the risk of obesity and diabetes [2,3], suggesting BCAA supplementation may be an effective dietary interevention to prevent obesity. In the current study, the acute isoleucine treatment had no beneficial effect on blood glucose levels in response to an OGTT. Further, chronic isoleucine supplementation was not an effective treatment for obesity and actually impaired glucose tolerance in SLD-mice.

In the current study, six weeks of chronic isoleucine treatment had no effect on body weight, gonadal fat pad mass, hepatic lipid content, and plasma lipid levels in SLD- and HFD-mice. Previously, four weeks of chronic isoleucine supplementation, protected mice from diet-induced weight gain and fat accumulation [7]. In that study, isoleucine supplementation was initiated after only two weeks of HFD feeding [7], which is arguably an insufficient time-course to establish obesity in mice [26]. Consistent with previous reports [27,28], 12 weeks of HFD feeding in the current study, led to significantly greater weight gain and adiposity compared to SLD-mice. Therefore, the findings suggest that isoleucine may be able to prevent diet-induced obesity, but may not be an effective treatment for reversing obesity.

Energy intake and energy expenditure are well-known effectors of body weight regulation [29]. In the current study, the chronic isoleucine treatment did not alter total energy expenditure, total energy intake, average RQ, or total activity in SLD- or HFD-mice. Previously, chronic isoleucine or leucine supplementation have been demonstrated to protect mice from diet-induced weight gain without reducing energy intake or increasing total activity levels [7,30]. Instead, leucine supplementation was observed to enhance total energy expenditure via an elevated 24 h oxygen consumption rate, normalised to body weight [30]. This elevated energy expenditure was attributed to an increased expression of uncoupling protein (UCP) 3 in thermogenic tissue [30]. In the isoleucine supplemented mice, energy expenditure was not directly measured [7]. However, expression of proteins involved in fatty acid uptake and oxidation were upregulated, namely FAT/CD36, PPARα and UCP2&3, in the liver and skeletal muscle [7]. This evidence suggests that chronic isoleucine or leucine supplementation in the previous reports may have protected from diet-induced fat gain through elevating fatty acid oxidation. However, in the current study, the chronic isoleucine treatment did not reduce 24 h RQ, suggesting no increase in lipid oxidation. Therefore, further investigation of lipid metabolism was not pursued. Indeed, the chronic supplementation of isoleucine in the current study, did not reverse the changes to energy expenditure observed in the HFD-induced obese mice.

In the current study, the chronic isoleucine treatment had no beneficial effect on fasting blood glucose levels in HFD-mice. This is consistent with a previous study where four weeks of isoleucine supplementation at a dose of 2.5%, did not affect fasting blood glucose levels in male mice fed a 45% HFD [7]. In contrast, six weeks of 2% chronic isoleucine supplementation significantly reduced fasting blood glucose levels compared to controls, in high-fat, high-sucrose fed female mice; a model of glucose intolerance [12]. These findings suggest that chronic isoleucine supplementation may be more effective in ameliorating fasting blood glucose elevated by high sucrose diets rather than high fat diets. In addition, there may be a difference in treatment effects between male and female mice. Future studies should, therefore, investigate the mechanisms of the effect of chronic isoleucine supplementation on glycaemic regulation, such as glucose uptake into tissues and hepatic glucose export. For example, acute administration of isoleucine to isolated myocytes stimulates the glucose uptake through enhanced recruitment of glucose transporters to the cell membrane [8]. Further, acute isoleucine administration in mice, supressed key enzymes in hepatic gluconeogenesis, namely phosphoenolpyruvate carboxykinase and glucose 6-phosphatase [31,32], contributing to lower fasting blood glucose levels. Therefore, it is possible that the effect of isoleucine on these blood glucose regulatory mechanisms persists under a chronic supplementation regime. However, there was no difference in blood glucose levels in the OGTT after acute isoleucine treatment in either SLD- or HFD-mice and, therefore, this was not pursued in the current study but may warrant future investigation.

In the current study, acute isoleucine orally administered at a dose of 0.3 g/kg, had no effect on postprandial blood glucose levels in SLD- or HFD-mice. Previously, acute administration of 0.3 g/kg isoleucine in lean [8] and diet-induced obese mice (60% HFD for eight weeks) [12] induced a dose-dependent reduction in postprandial blood glucose levels. Further, acute isoleucine administration in obese leptin receptor-deficient (*db/db)* mice, significantly reduced blood glucose levels in response to an OGTT, albeit at a higher dose of 0.5 g/kg [12]. Considering this observed dose-dependent effect, a larger dose may have been necessary to reduce the blood glucose AUC in the current more chronic (12 weeks) model of diet-induced obesity. However, this does not explain why no effect was observed in the SLD-mice.

Consistent with a previous report [12], the chronic isoleucine treatment had no beneficial effect on blood glucose levels in response to an OGTT in HFD-mice. In contrast, the chronic isoleucine supplementation impaired glucose tolerance in lean mice, suggesting an adverse effect of chronic isoleucine supplementation. It has been reported that a western diet low in BCAAs reduced the blood glucose AUC compared to mice fed a standard western diet [33]. Furthermore, consistent with the current study a western diet supplemented with BCAAs had no effect on blood glucose AUC compared to mice fed a standard western diet [33]. Further, there is some evidence that obesity-induced BCAA dysmetabolism may promote insulin resistance through mitochondria dysfunction and impaired fatty acid oxidation [34,35]. Whether chronic isoleucine supplementation induces BCAA dysmetabolism, and promotes insulin resistance in lean mice, requires further investigation.

Slowing gastric emptying allows for efficient digestion and absorption of nutrients, including glucose [13,36,37]. In the current study, the acute and chronic isoleucine treatment did not affect gastric emptying in SLD-mice. Further, there was no significant correlation between postprandial blood glucose levels and gastric emptying in any groups. Previously, the consumption of whey protein, rich in BCAAs, including isoleucine [38], before a meal has been demonstrated to slow gastric emptying and reduce postprandial hyperglycaemia in people with type 2 diabetes [14,15]. Similarly, intragastric adminsitartion of isoleucine slowed gastric emptying and reduced postprandial glucose levels in lean participants [16]. However, in rodent studies, the effect of isoleucine administration on gastric emptying is less clear. For example, in rats, oral administration of isoleucine reduced the ^13^CO_2_ content in the breath following a gastric emptying breath test, but did not significantly affect the ^13^CO_2_ AUC, Cmax, or Tmax values (peak concentration and the time at which it occurred) compared to controls [39].

## 5. Conclusions

The acute and chronic isoleucine supplementation had no beneficial effect to limit HFD-induced body weight gain, adiposity, fasting blood glucose levels, and glucose tolerance. In contrast, the chronic isoleucine treatment impaired glucose tolerance in SLD-mice. Therefore, the chronic isoleucine supplementation is unlikely to be an effective dietary intervention for the treatment of obesity and type 2 diabetes.

## Figures and Tables

**Figure 1 nutrients-12-02446-f001:**
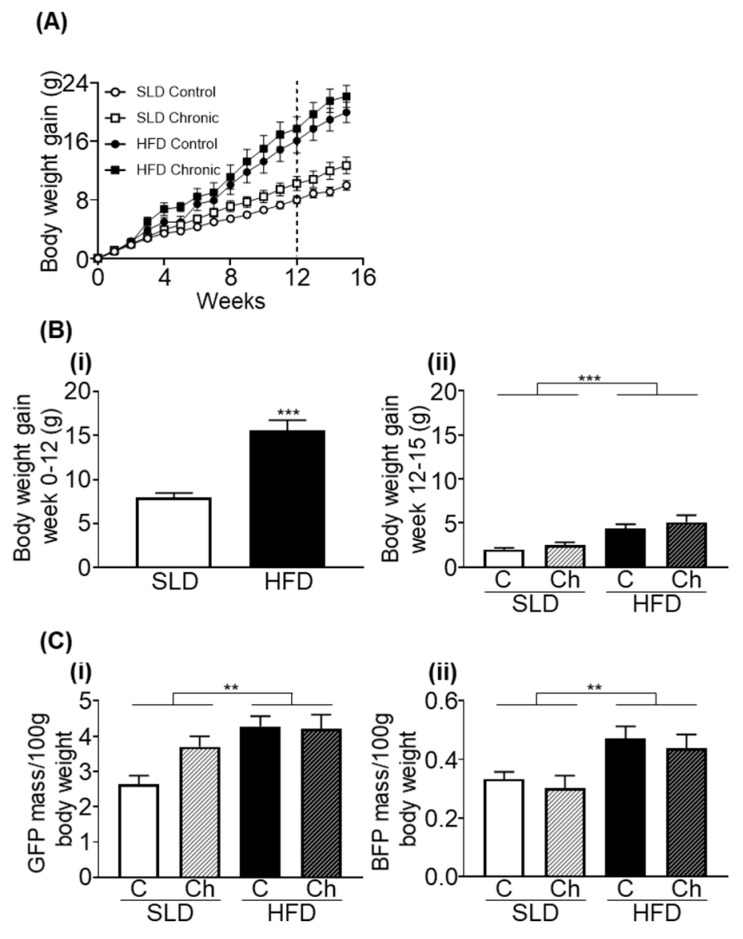
High fat diet (HFD) feeding but not the chronic isoleucine treatment increased weight gain and adiposity in mice. (**A**) Body weight gained during weeks 0–15 in mice fed a standard laboratory diet (SLD) or HFD (SLD/HFD-Control (C; control and acute groups pooled as no acute gavage of isoleucine had occurred at this point) *n* = 16–20; SLD/HFD-Chronic (Ch) *n* = 8–10). The dotted line indicates the onset of chronic isoleucine supplementation. (**B**) (**i**) Body weight gained during week 0–12 in SLD or HFD (*n* = 24–30/group; all HFD and SLD groups pooled as no chronic or acute isoleucine treatment had occurred at this point); *** *p* < 0.001 unpaired t-test. (**ii**) Body weight gained in week 12–15 (SLD/HFD-Control (C; control and acute groups pooled as no acute gavage of isoleucine had occurred at this point) *n* = 16–20; SLD/HFD-Chronic (Ch) *n* = 8–10). (**C**) (**i**) Gonadal fat pad (GFP) mass and (**ii**) brown fat pad (BFP) mass per 100 g of total body weight (*n* = 8–10/group). Values are mean ± SEM. ** *p* < 0.01, *** *p* < 0.001; diet effect, two-way ANOVA.

**Figure 2 nutrients-12-02446-f002:**
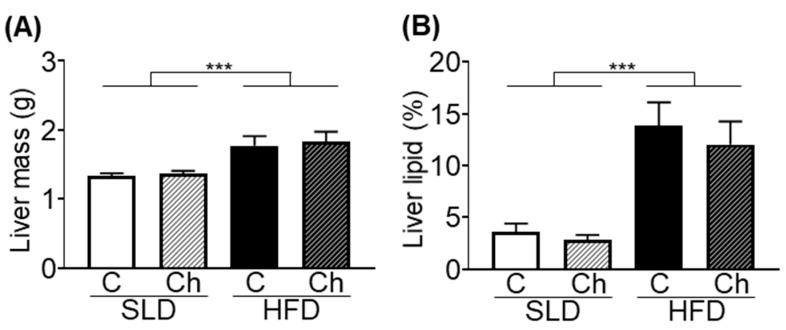
High fat diet (HFD) feeding but not chronic isoleucine treatment increased the liver mass and lipid content in mice. (**A**) Liver mass and (**B**) percentage lipid area per 1 mm^2^ liver area of standard laboratory diet (SLD) and HFD-mice (SLD/HFD-Control (C) and SLD/HFD-Chronic (Ch), *n* = 8–10). Values are mean ± SEM. *** *p* < 0.001; diet effect, two-way ANOVA.

**Figure 3 nutrients-12-02446-f003:**
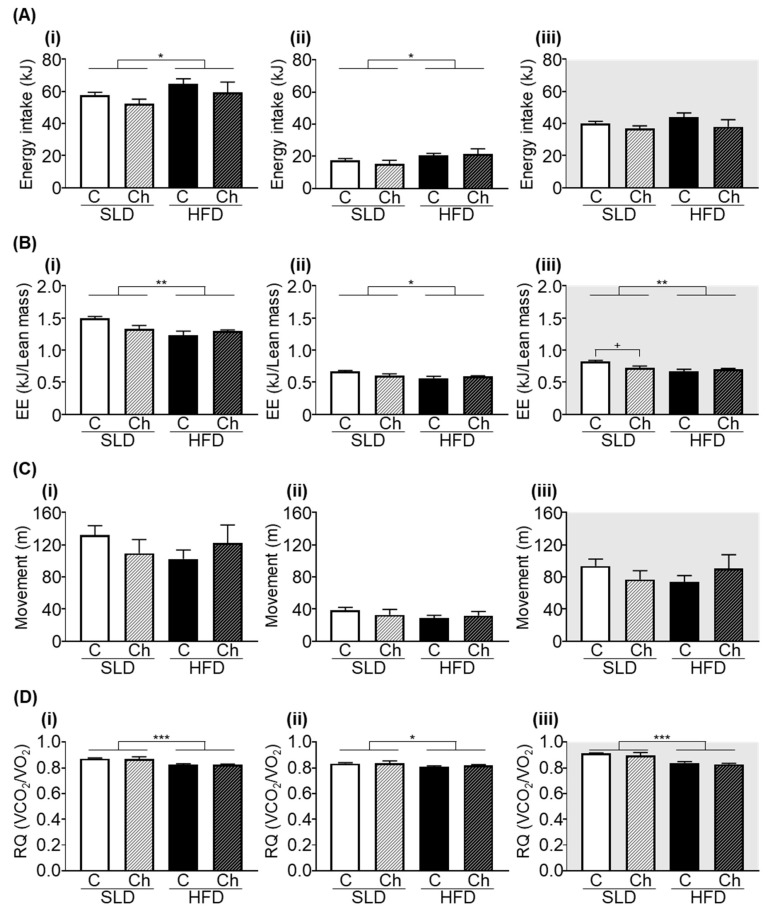
High fat diet (HFD) feeding but not the chronic isoleucine treatment affects energy balance in mice. (**A**) Energy intake, (**B**) energy expenditure (EE), (**C**) activity (distance of movement), and (**D**) respiratory quotients (RQ), across 24 h (**i**), 12 h of day (light phase) (**ii**), and 12 h of night (dark phase) (**iii**) in a standard laboratory diet (SLD) and HFD-mice (SLD/HFD-Control (C) *n* = 16–20, SLD/HFD-Chronic (Ch) *n* = 8–10). Values are mean ± SEM. * *p* < 0.05, ** *p* < 0.01, *** *p* < 0.001, diet effect; two-way ANOVA with Sidak’s *post hoc* test, +*p* < 0.05.

**Figure 4 nutrients-12-02446-f004:**
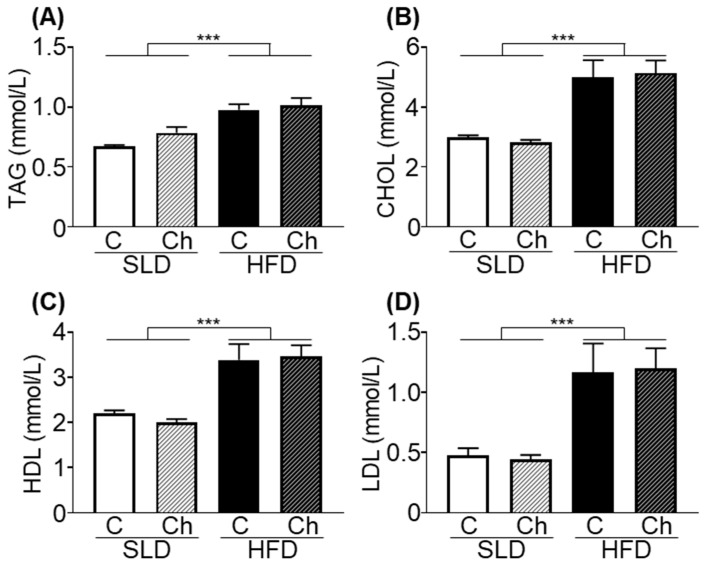
High fat diet (HFD) feeding but not the chronic isoleucine treatment elevated plasma lipid metabolites in mice. Plasma (**A**) total triglycerides (TAG), (**B**) total cholesterol (CHOL), (**C**) high density lipoprotein (HDL), and (**D**) low density lipoprotein (LDL) levels in a standard laboratory diet (SLD) and HFD-mice (SLD/HFD-Control (C) *n* = 8–10, SLD/HFD-Chronic (Ch) *n* = 8–10). Values are mean ± SEM *** *p* < 0.001, diet effect; two-way ANOVA.

**Figure 5 nutrients-12-02446-f005:**
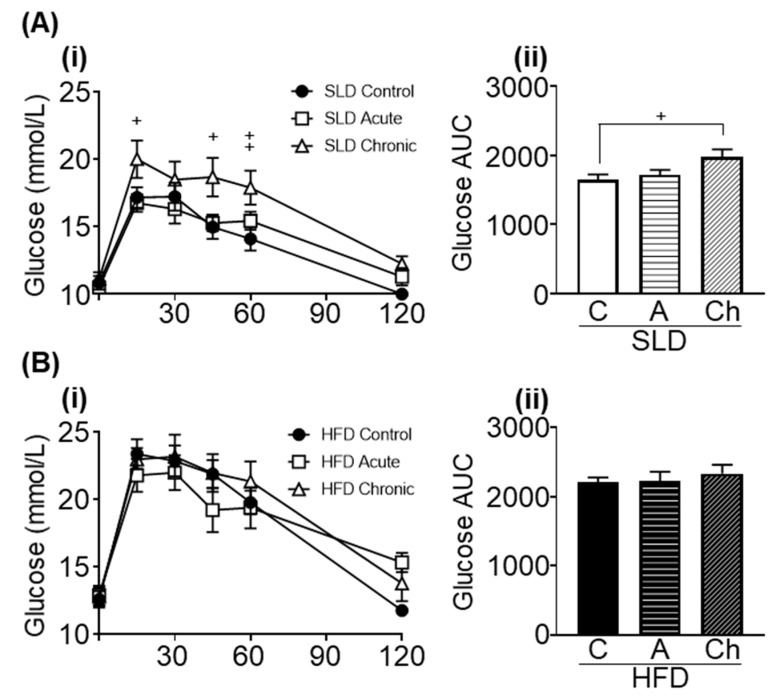
Acute (A) and chronic (Ch) isoleucine treatment differentially affects glucose tolerance. (**i**) Blood glucose levels in response to an oral glucose tolerance test and (**ii**) glucose area under curve (AUC) in (**A**) standard laboratory diet (SLD) and (**B**) high fat diet (HFD)-mice. (SLD/HFD-Control (C) *n* = 8–10, SLD/HFD-A *n* = 8–10 and SLD/HFD-Ch *n* = 8–10). Values are mean ± SEM. + *p* < 0.05, one-way ANOVA.

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
