# Peer review of "The Effect of Isoleucine Supplementation on Body Weight Gain and Blood Glucose Response in Lean and Obese Mice"

_nutrients, 2020, doi:10.3390/nu12082446_

Round 1

Reviewer 1 Report

Problem

The manuscript presents findings of isoleucine supplementation (chronic and acute)in lean (standard laboratory diet; study controls) and obese mice(obesity induced by high fat diet). After 12 weeks of their respective diets, animals were randomized into subgroups with treated animals given isoleucine supplementation for 6weeks. At week 17, the acute effect was measured. The following endpointswere considered:body weight,glucose tolerance(OGGTand glucoseAUC), energy expenditure, adiposity.

General aspects

-Minor revisions needed

-Overall, well-designedand presented study

Title

-Title does not include finding(s)of the study or information about the study itself; the effect of ...is very general. Consider changing to Acute and chronic isoleucine supplementation...instead.

Methods

-In the Discussion section(#284-286), the authors suggested that chronic isoleucine supplementation may be more effective in ameliorating fasting blood glucose elevated by high sucrose diets rather than high fat diets. Why was the high-fat diet chosen in the study design?

Introduction

-Study aim presented in the abstract (#28)refers to body weight and glucose tolerance”, but the study aim in the Introduction(#70)is presented as, e.g.“alters body weight gain. However, study hypothesis (#66-68) only mentions improvement in glucose metabolism.Add the expected effect of supplementation on body weight in the hypothesis.

Results

-#152: Authors did not hypothesize that isoleucine treatment affects weight gain. Conversely, they provide evidence to support protective effect against fat gain. -#176 and 186:The statement HFD feeding, but not chronic isoleucine treatmentprovides the impression that HFD feeding is the focus of the study, not isoleucine treatment.

Discussion

-#292-294:Were the levels of those enzymes measured in this study? If no, why?

-#299-300: Considering those arguments and the final conclusion (#327-328) would the use of db/db mice be a better model for this type of experiment?

-#315-317: Consider adding information linking BCAA with whey protein such as Whey protein, which has the highest isoleucine content...(ref)

Figures

-Figure 5. (A) and (B): Symbols on curves for all three lines are overlapping. An adjustment of the Y-axis scale would be advisable.

Author Response

Comment: Title does not include finding(s)of the study or information about the study itself; “the effect of ...”is very general. Consider changing to “Acute and chronic isoleucine supplementation...”instead.

Response: We have considered the reviewer’s suggestion, but have decided to keep the original title, as we believe that it provides readers with a better sense of the purpose of the study.

Comment: In the Discussion section (#284-286), the authors suggested that “chronic isoleucine supplementation may be more effective in ameliorating fasting blood glucose elevated by high sucrose diets rather than high fat diets”. Why was the high-fat diet chosen in the study design?

Response: We chose the high-fat diet based on existing use of the high-fat diet-induced obese mouse model for obesity research. Consistent with previous literature, high-fat diet feeding in C57BL/6 mice induces obesity (Lang, et al. Scientific Reports 9, no. 1 (2019): 19556) and impairs glucose tolerance (Winzell, Maria Sörhede, and Bo Ahrén. Diabetes 53, no. suppl 3 (2004): S215-S19.) and, therefore, we considered it an appropriate model. As outlined in the discussion, a high-fat, high-sucrose diet may have differentially affected the response to the isoleucine supplementation. This hypothesis warrants evaluation in future studies. We included a comment to this effect in the Discussion (lines 287-294): “In contrast, 6 weeks of 2% chronic isoleucine supplementation significantly reduced fasting blood glucose levels compared to controls, in high-fat, high-sucrose fed female mice; a model of glucose intolerance [12]. These findings suggest chronic isoleucine supplementation may be more effective in ameliorating fasting blood glucose elevated by high sucrose diets rather than high fat diets. In addition, there may be a difference in treatment effects between male and female mice. Future studies should, therefore, investigate the mechanisms of the effect of chronic isoleucine supplementation on glycaemic regulation, such as glucose uptake into tissues and hepatic glucose export.

Comment: Study aim presented in the abstract (#28) refers to “body weight and glucose tolerance”, but the study aim in the Introduction(#70)is presented as, e.g.“alters body weight gain”. However, study hypothesis (#66-68) only mentions improvement in glucose metabolism. Add the expected effect of supplementation on body weight in the hypothesis.

Response: Thank you for pointing out these inconsistencies. We have rectified these issues by changing the text as follows:

Abstract, line 29: This study aimed to investigate the impact of chronic isoleucine on body weight gain and glucose tolerance in lean and high-fat-diet (HFD) induced-obese mice.”

Hypothesis, lines 66-68: Therefore, we hypothesised that chronic isoleucine supplementation slows gastric emptying, improves glucose tolerance and reduces body weight gain in mice.”

Comment: Authors did not hypothesize that isoleucine treatment affects weight gain. Conversely, they provide evidence to support protective effect against fat gain.

Response: We have corrected the wording in the hypothesis (line 29): This study aimed to investigate the impact of chronic isoleucine on body weight gain and glucose tolerance in lean and high-fat-diet (HFD) induced-obese mice.”

Comment: #176 and 186: The statement “HFD feeding, but not chronic isoleucinetreatment” provides the impression that HFD feeding is the focus of the study, not isoleucine treatment.

Response: These subheadings have been changed accordingly to:

Line 176: “Chronic isoleucine treatment does not affect liver lipid content.”

Line 186: “Chronic isoleucine treatment does not alter energy intake, energy expenditure, activity and respiratory quotient.”

Line 214: “Chronic isoleucine treatment does not affect plasma lipid metabolites.”

Comment: #292-294: Were the levels of those enzymes measured in this study? If no, why?

Response: These enzymes were not measured, especially considering we did not see a difference in blood glucose levels after acute isoleucine in the OGTT in both SLD and HFD- mice. We have now added a sentence to the discussion explaining this (lines 300-302): However, there was no difference in blood glucose levels in the OGTT after acute isoleucine treatment in either SLD or HFD-mice and, therefore, this was not pursued in the current study but may warrant future investigation.”

Comment: #299-300: Considering those arguments and the final conclusion (#327-328) would the use of db/db mice be a better model for this type of experiment?

Response: The acute effect of isoleucine in the db/db mice was at a much higher dose than that observed in the previous HFD study mentioned in the discussion (Ikehara et al. Biol Pharm Bull 31, no. 3 (2008): 469-72). Therefore, we believe the db/db mice would not be a better model for this type of experiment.

Comment: #315-317: Consider adding information linking BCAA with whey protein such as “Whey protein, which has the highest isoleucine content...(ref)”

Response: We have changed this section accordingly (line 325-327): “Previously, the consumption of whey protein, rich in BCAAs, including isoleucine [38], before a meal has been demonstrated to slow gastric emptying and reduce postprandial hyperglycaemia in people with type 2 diabetes [14, 15].”

Comment: Figure 5. (A) and (B): Symbols on curves for all three lines are overlapping. An adjustment of the Y-axis scale would be advisable.

Response: We have changed the graphs figure 5 accordingly, with the axis now from 10-25 instead of 8-28. There is still a degree of overlap, but this is unavoidable, simply because some of the data are not statistically different.

Reviewer 2 Report

This interesting work depicts the effects of chronic isoleucine supplementation dietary may not be an effective treatment for obesity or glucose intolerance. Although I find that the work is well justified and introduced, there are a few points that need to be addressed.

The topic seems to be described before by N. E. Cummings and others (2018) that the consumption of a Western diet reduced in BCAAs was also accompanied by a dramatic improvement in glucose tolerance and insulin resistance, for that suggest that specifically reducing dietary BCAAs may represent a highly translatable option for the treatment of obesity and insulin resistance.

On the other hand, the authors should clearly show the sweep of the body weight gain for each group and present on a graph only for 1-16 weeks: SLD-C; SLD-Ch; HFD-C and HFD-Ch.

Respect to Oral Glucose Tolerance Test, the authors suggest six hours of fast for the mice before receiving an oral gavage of either isoleucine and I consider that I should inspect and modify with 14 hours and administration with 2g / Kg BW of glucose.

I consider that the authors should present the weight of tissue (GFP and BFP) about 100 g of body weight of animals

Author Response

Comment: The topic seems to be described before by N. E. Cummings and others (2018) that the consumption of a Western diet reduced in BCAAs was also accompanied by a dramatic improvement in glucose tolerance and insulin resistance, for that suggest that specifically reducing dietary BCAAs may represent a highly translatable option for the treatment of obesity and insulin resistance.

Response: Thank you for pointing out this paper. We have now added this to the manuscript (Lines 307-311): “It has been reported that a western diet low in BCAAs reduced the blood glucose AUC compared to mice fed a standard western diet [33]. Furthermore, consistent with the current study a western diet supplemented with BCAAs had no effect on blood glucose AUC compare to mice fed a standard western diet [33].”

Comment: On the other hand, the authors should clearly show the sweep of the body weight gain for each group and present on a graph only for 1-16 weeks: SLD-C; SLD-Ch; HFD-C and HFD-Ch.

Response: We have now added the body weight gain data to figure 1. While producing this figure we noticed that the weight of the mice just prior to commencing the diet was recorded as week 1 when in fact it should have been the 0 timepoint. We have corrected this error in the text.

Comment: Respect to Oral Glucose Tolerance Test, the authors suggest six hours of fast for the mice before receiving an oral gavage of either isoleucine and I consider that I should inspect and modify with 14 hours and administration with 2g / Kg BW of glucose.

Response: We have chosen a six-hour fast based on the fact that this is sufficient time to empty the stomach without causing excessive stress. It is a common fasting period prior to a GTT (e.g. Andrikopoulos, S., et al. Am J Physiol Endocrinol Metab 295 (6) (2008): E1323-32). We used a lower dose of glucose as it is common to have a hyperglycemic response in the high-fat diet-induced obese mice. To eliminate this possibility it is now considered appropriate to use a lower concentration of glucose in high-fat diet studies (Nagy, C., and E. Einwallner. J Vis Exp, no. 131 (2018)). We have now added this information to the methods section (lines 98-102): “Consistent with previous studies [19, 20], mice were fasted for six hours (0800 - 1400 hr) before receiving an oral gavage of either isoleucine (SLD/ HFD-A groups) or water (SLD/ HFD-C and SLD/ HFD-Ch groups). After 30 mins, all mice received an oral gavage of 20% D-glucose (1 g/ kg BW), a dose chosen to ensure the HFD-mice did not experience a severe hyperglycemic response with blood glucose levels beyond the range of the glucose monitor.

Comment: I consider that the authors should present the weight of tissue (GFP and BFP) about 100 g of body weight of animals

Response: We are not entirely certain that we understand the reviewer’s comment, however, we have now expressed the tissue weight (Figure 1) relative to 100g body weight of animals, which is the same as percentage body weight (weight of GFP or BFP / total body weight * 100).

Reviewer 3 Report

The authors of the paper entitled "The effect of dietary isoleucine supplementation on 2 body weight and blood glucose response in lean and 3 obese mice" write that their study is up-to-date. However, they base their work on publications from 1984 to 2018. Most of the literature is from over 5 years. One should refer to the latest research (up to 5 years). The number of the Bioethics Committee should also be provided.

Author Response

Comment: The authors of the paper entitled "The effect of dietary isoleucine supplementation on 2 body weight and blood glucose response in lean and 3 obese mice" write that their study is up-to-date. However, they base their work on publications from 1984 to 2018. Most of the literature is from over 5 years. One should refer to the latest research (up to 5 years).

Response: We appreciate the reviewer’s comment and confirm that the publications cited reflect the literature in this field. However, we have rechecked the current literature and added the following publications:

Cummings, N. E., et al. J Physiol 596, no. 4 (2018): 623-45.

Nie, C., T. et al. Int J Mol Sci 19, no. 4 (2018): 954.

Holecek, M. Nutr Metab (Lond) 15 (2018): 33.

Comment: The number of the Bioethics Committee should also be provided.

Response: The approval details were stated in the methods section, with the ethics approval number added after submission on the request of the journal (Page 2, Line 74-76): This study was approved (Ethics approval: SAM237) by the South Australian Health and Medical Research Institute Animal Ethics Committee. All experimental protocols were performed in alignment with the Australian Code of Practice for the Care and Use of Animals for Scientific Purposes.”

Round 2

Reviewer 2 Report

I accept the modifications made by authors to finish the review and it can be submitted.

Reviewer 3 Report

no other comments